# Decrease in the Size of Fat-Enlarged Axillary Lymph Nodes and Serum Lipids after Bariatric Surgery

**DOI:** 10.3390/cells11030482

**Published:** 2022-01-30

**Authors:** Dennis Dwan, Seth K. Ramin, Youdinghuan Chen, Kristen E. Muller, Roberta M. diFlorio-Alexander

**Affiliations:** 1Department of Radiology, Beth Israel Deaconess Medical Center, 330 Brookline Ave, Boston, MA 02215, USA; ddwan@bidmc.harvard.edu; 2Department of Medicine, Geisel School of Medicine at Dartmouth, Hanover, NH 03755, USA; Seth.K.Ramin.MED@dartmouth.edu; 3Faculty of Science, Wilmington University, 320 N Dupont Hwy, New Castle, DE 19720, USA; ydavidkchen@gmail.com; 4Department of Pathology, Dartmouth-Hitchcock Medical Center, 1 Medical Center Drive, Lebanon, NH 03766, USA; Kristen.E.Muller@hitchcock.org; 5Department of Radiology, Dartmouth-Hitchcock Medical Center, 1 Medical Center Drive, Lebanon, NH 03766, USA

**Keywords:** fat enlarged lymph nodes, obesity, bariatric surgery, screening mammography, lymphatic dysfunction, dyslipidemia

## Abstract

Background: Ectopic fat deposition in obesity is associated with organ dysfunction; however, little is known about fat deposition within the lymphatic system and associated lymphatic dysfunction. Methods: One hundred fifty-five women who underwent routine screening mammography before and after a Roux-en-y gastric bypass or a sleeve gastrectomy were retrospectively reviewed and after excluding women without visible nodes both before and after bariatric surgery, 84 patients were included in the final analysis. Axillary lymph node size, patient weight in kilograms, body mass index, and a diagnosis of hypertension, type 2 diabetes, and dyslipidemia were evaluated before and after surgery. Binary linear regression models and Fischer’s exact test were used to evaluate the relationship between the size of fat-infiltrated axillary lymph nodes, patient age, change in patient weight, and diagnosis of hypertension, type 2 diabetes, and dyslipidemia. Results: Fat-infiltrated axillary lymph nodes demonstrated a statistically significant decrease in size after bariatric surgery with a mean decrease of 4.23 mm (95% CI: 3.23 to 5.2, *p* < 0.001). The resolution of dyslipidemia was associated with a decrease in lymph node size independent of weight loss (*p* = 0.006). Conclusions: Mammographically visualized fat-infiltrated axillary lymph nodes demonstrated a statistically significant decrease in size after bariatric surgery. The decrease in lymph node size was significantly associated with the resolution of dyslipidemia, independent of weight loss, age, and type of surgery.

## 1. Introduction

The incidence of obesity is steadily increasing with current data estimating that over 30% of adults worldwide are obese [1]. Excess adiposity is closely linked with an increased risk of cardio-metabolic disease, the leading cause of mortality in the Western world. However, there is marked heterogeneity in the obesity phenotype that is not explained by body mass index (BMI). This heterogeneity may explain why approximately 10–30% of people with obesity, defined by BMI, do not manifest cardio-metabolic disease. This phenotype of “metabolically healthy obesity” is associated with lower ectopic fat within and around organs. This is consistent with evidence demonstrating that the distribution of excess fat throughout the body is more closely correlated with cardiometabolic disease and adverse health outcomes than BMI [2,3]. Specifically, the obesity phenotype is better characterized by ectopic fat deposition within tissues that are not physiologically designed for adipose storage, and that normally contain only small amounts of adipose tissue, such as the visceral cavity, liver, heart, and skeletal muscle [4]. It has been proposed that in this era of precision medicine, cardiovascular risk assessment among the obese should be refined by measures of ectopic fat deposition rather than BMI [5]. In addition, improved cardiometabolic function is more strongly associated with decreased ectopic fat than decreased BMI, and ectopic fat is therefore considered a biomarker that should be monitored with weight loss intervention [6]. The reversal of cardiometabolic disease with decreased ectopic adipose stores in and around organs is felt to reflect local and systemic effects of improved organ function secondary to a decrease in inflammation and dysregulated lipid metabolism, as well as a reversal of structural changes associated with ectopic fat [6].

Despite ongoing research evaluating ectopic fat deposition throughout the body, there has been very limited evaluation of ectopic adipose tissue within the lymphatic system. Fat-enlarged lymph nodes are considered a benign variant of nodal morphology in contrast to structural changes associated with nodal hyperplasia or metastatic adenopathy. Adipose tissue, once felt to be inert, is now known to function as a paracrine organ that secretes hormones and adipokines. Ectopic adipose tissue in the liver, kidney, visceral cavity, heart, and muscle is dysfunctional compared to classic physiologic subcutaneous adipose tissue due to altered adipokine secretion and inflammation [6]. Our hypothesis is that ectopic lymph node fat may have adverse effects, similar to changes described with other ectopic fatty depots [3,4,5,6]. Recent reports from our group and others have demonstrated a relationship between fat-infiltrated lymph nodes and obesity [7,8]. We have also demonstrated an association between fatty nodes and non-alcoholic fatty liver disease and we most recently showed that fatty nodes were associated with a higher likelihood of axillary metastases in obese women with breast cancer independent of patient and tumor characteristics, including BMI [9,10].

Fat-infiltrated lymph nodes identified on imaging studies correspond to an enlargement of the central radiolucent fatty hilum. There is no associated increase in the size of the radiodense peripheral functional cortex as is seen with reactive and malignant adenopathy. In contrast, the cortex is often thin and effaced. The morphologic changes of fat-infiltrated nodes are visualized on mammograms, MRI, and histology as shown in Figure 1 and Figure 2. While fatty lymph nodes are reported to be benign, their significance relative to cardio-metabolic disease is unknown. Furthermore, it is unknown if the amount of fat deposition within the nodal hilum is reversible with weight loss and associated with improved cardiometabolic disease status as has been shown in other organs affected by ectopic lipid accumulation. Therefore, we designed this retrospective study to evaluate nodal fat deposition and metabolic disease in obese patients after bariatric surgery. Specifically, the aims of this study were to: (1) evaluate the association between weight loss and change in size of mammographically visualized fat-infiltrated axillary lymph nodes and (2) evaluate the association between the decrease in the size of fat-enlarged nodes and the resolution of metabolic disease.

## 2. Materials and Methods

### 2.1. Study Population

The present retrospective cohort study was approved by the Institutional Review Board (IRB) at our institution, and consent for this retrospective review was waived. The initial study population included 731 women aged 40 years or older who had bariatric surgery between 2011 and 2018. Requirements for bariatric surgery eligibility at our institution include a body mass index (BMI) greater than 40 kg/m^2^ or a BMI between 35 and 40 kg/m^2^ in a patient with obesity-related comorbidities. Women who did not have a screening mammogram performed before and at least 6 months after bariatric surgery were excluded, resulting in 168 patients. Patients with mammograms performed at outside institutions that were not available for review were also excluded, resulting in a final sample size of 155 patients. All patients in the final cohort were white, consistent with the predominantly white patient population in our rural catchment area.

### 2.2. Methodology

Patient data were collected by reviewing the electronic medical record (Epic Systems Corporation, Verona, WI, USA). Two-dimensional full field digital mammograms were obtained with Selenia and Dimensions hologic units (Hologic Incorporation, Bedford, MA). Images were reviewed on Barco 3-megapixel MDCG-3221 monitors (Kotrijk, Belgium) with Philips PACS v. 3.6 (Philips Healthcare; Best, Netherlands). Independent lymph node measurements were obtained by a breast radiologist with 18 years of experience (RDA) and a medical student (DD), who were both blinded to patient history. The medial lateral oblique (MLO) views were evaluated for each patient and the largest and most clearly visualized axillary lymph node from either MLO view was chosen as the index node for measurement before and after bariatric surgery. Lymph node length was obtained along the longest visible axis of the node as shown in Figure 3.

### 2.3. Clinical Definitions

Clinical characteristics of hypertension, type 2 diabetes (T2D), and dyslipidemia were recorded as binary variables based on whether a diagnosis was present or absent in the patient’s electronic medical record. Hypertension was defined per the Eighth Joint National Committee criteria as a blood pressure greater than 150/90 mmHg in a patient 60 years of age or older or a blood pressure greater than 140/90 mmHg in a patient younger than 60 years of age [11]. A diagnosis of T2D was based on the American Diabetes Association definition of a hemoglobin A1C ≥ 6.5% or fasting glucose ≥ 126 mg/dL [12]. Dyslipidemia was defined according to the most recent up-to-date clinical guidelines as a total cholesterol > 200 mg/dL, LDL > 100 mg/dL, HDL < 40 mg/dL, triglycerides > 150 mg/dL, or a combination thereof [13]. A fat-infiltrated lymph node was defined as a node demonstrating an increased radiolucent fatty hilum relative to radio-dense peripheral cortex measuring greater than 14 mm in length, as identified on our prior study evaluating fat-infiltrated axillary lymph nodes on digital mammography [7]. Axillary lymph nodes were considered to demonstrate a change in size if measurements differed by more than 3.5 mm between pre- and post- bariatric surgery mammograms. The rationale for determining this threshold was to account for inter- and intra-rater variation in lymph node measurements. This threshold was determined by employing a data-driven approach. Briefly, thresholds ranging from −8.7 to 22.8 at increments of 0.1 were explored, and the median specificity and median sensitivity for each threshold was calculated by using RDA’s measurements as a gold standard compared to DD’s. A threshold of 3.5 mm yielded a sensitivity of 1.00 and specificity of 0.98 (optimal combination) and was chosen to represent the ground-truth change in lymph node size.

### 2.4. Statistical Analysis

Inter-rater reliability between the two investigators was assessed by Pearson’s correlation analysis. Assuming a pooled standard deviation of 8.4 and per-group sample sizes of 39 and 45, there was a 93% power to detect a 6.4-mm change in LN at the *p*-value threshold of 0.05. Lymph node measurements for an agreed-upon index node between the two investigators demonstrated a very strong inter-rater agreement (Pearson’s correlation coefficient = 0.949).

A paired *t*-test was used to compare the mean size between patients that had no change in nodal size to patients with a change in lymph node size after bariatric surgery. A multivariate logistic regression model was used to evaluate the relationship between change in hypertension, T2D, and dyslipidemia and a change in lymph node size adjusted for weight, age, and type of surgery. A multivariate linear regression was used to evaluate for weight loss adjusted by age, initial weight, and presence of T2D. All statistical analyses were performed in R (v. 3.5.1) and IBM SPSS (v. 24).

## 3. Results

Due to normal variation in patient positioning and changes in body habitus with weight loss, axillary nodes in the mediolateral oblique view were variably visualized on screening mammograms (Figure 3). Of 155 obese patients with bariatric surgery who had screening mammograms before and after surgery, 21 (13.5%) were excluded because no mammographic lymph nodes were visualized in the axilla, and 10 (6.5%) were excluded because visible nodes appeared normal and did not meet the size and morphology criteria for fat-infiltrated lymph nodes. Of the remaining 125 patients, 40 patients were excluded because they did not have visible fat-infiltrated nodes on both pre- and post-bypass surgery mammograms (Appendix A). The remaining 84 patients (54.8%) had measurable fat-enlarged nodes visualized both before and after bariatric surgery and were used for the analysis. Thirty-nine (25.1%) patients had no change in lymph node size and 45 (29%) patients had a change in size per the threshold described in the methodology.

When directly comparing patients with a change in FIN size and patients without a change in FIN size after bariatric surgery, there was no significant difference in age, BMI, or prevalence of metabolic disease such as diabetes, dyslipidemia, or hypertension (Table 1). There was also no significant difference in the type of anti-diabetic or anti-lipid medication prescribed before bariatric surgery. Only a Roux-en-y gastric bypass or a sleeve gastrectomy were performed in this cohort and either surgery was equally represented among the two groups. Average time from bariatric surgery to the selected post-bypass mammogram assessed for FIN size change was not significantly different between the two groups (22.8 months in the group demonstrating no change and 21.8 months in the group demonstrating change in FIN size, *p* = 0.77). A direct comparison of lymph node size between patients with and without hypertension, diabetes, and dyslipidemia prior to bariatric surgery was performed (Table 2). Although the size of FINs was greater in patients with metabolic disease, the mean difference in size was not significant (3.3 mm, *p* = 0.39).

A multivariate linear regression model analyzing weight loss adjusted for age, presence of T2D, and initial weight was performed. The fitted regression model was weight loss in kilograms = −12.1 − 0.17 *(age) + 0.02 *(presence of T2D) + 0.45 *(initial weight in kg). Only initial weight significantly predicted weight loss in this model (β = 0.4, *p* < 0.01); age (β = −0.17, *p* = 0.37) and the presence of T2D (β = 0.02, *p* = 0.2) did not.

After bariatric surgery, fat-enlarged nodes had a mean decrease in size of 4.23 mm (95% CI: 3.23 to 5.2, *p* < 0.001, two-tailed paired *t*-test). The resolution of dyslipidemia was found to be significantly associated with a decrease in lymph node size (OR = 5.5, 95%CI = 1.6–18.7, *p* = 0.006), independent of weight loss or age. Change in hypertension or diabetes was not found to be statistically associated with a change in weight or lymph node size (Table 3).

## 4. Discussion

BMI is a simple anthropometric index of adiposity that does not accurately reflect the risk of cardiometabolic disease among the obese. As such, BMI is not included as a component of the Framingham CV risk assessment tool [14], and a recent position statement from the International Atherosclerosis Society has underscored the need to develop tools to assess changes in ectopic fat to better reflect cardio-metabolic risk [5]. Lymph nodes do not represent a classic adipose tissue storage depot, and normal nodes contain only a small amount of fat within the lymph node hilum. Therefore, lymph nodes enlarged by hilar adipose deposition likely represent a novel ectopic fatty depot. Our study showed that fat-expanded axillary lymph nodes may decrease in size after weight loss, and that a decrease in fatty lymph node size was associated with the resolution of dyslipidemia independent of patient age and weight loss (*p* = 0.006). This is the first study to show a direct relationship between improved cardiometabolic disease status and decreased ectopic fat deposition in the immune–lymphatic system. These findings are concordant with reports of improved cardio-metabolic disease associated with decreased liver, muscle, cardiac, and pancreatic ectopic fat after weight reduction [15]. Lipid parameters improved in patients with and without a decrease in fatty lymph node size in our study. However, the mean LDL normalized in patients with a decrease in fatty node size resulting in the discontinuation of statin therapy while mean LDL remained abnormal in patients who did not have a decrease in the size of fat-infiltrated nodes. LDL is the most clinically relevant serum lipid marker and studies have consistently demonstrated an association between elevated LDL and risk of mortality from CV disease. Studies have further demonstrated a decrease in the risk of mortality with LDL reduction in patients with and without CVD [16,17,18]. Serum LDL value is the focal point of treatment and drives the clinical decision to initiate statin therapy; therefore, optimizing LDL levels is the main goal of current guidelines, including the American Heart Association and the American College of Cardiology [19]. Our results also showed a significant association between decreased fatty node size and fasting glucose as well as the resolution of T2DM; however, these associations did not remain significant after adjusting for patient age and BMI. Weight reduction surgery resulted in improvements in HTN, T2DM, and dyslipidemia in patients with and without a decrease in the size of fatty axillary lymph nodes. However, the resolution of dyslipidemia was the only condition that was significantly associated with a decrease in the size of fat-infiltrated axillary nodes after bariatric surgery that remained significant after adjusting for patient age and change in BMI. Our results suggest that hilar nodal fat may represent a novel ectopic fatty depot and that decreased ectopic nodal adipose after weight loss may be associated with the resolution of dyslipidemia, findings that are concordant with prior large-scale studies demonstrating the reversal of cardiometabolic disease with decreased ectopic fat.

The study of ectopic fat deposition has been advanced with the use of imaging techniques such as CT and MRI that are able to quantify fat within and around organs. Our study leverages the visibility of axillary nodes on screening mammograms; however, axillary nodes are not visualized in up to 50% of mammograms compared to the near complete visualization of axillary nodes on MRI and CT [20]. Variable axillary node visualization on mammography is due to the deep supero-lateral anatomic location of level I axillary nodes, differences in patient body habitus, and variable positioning techniques that allow only partial visualization of the deep lymph nodes in the axilla. Although axillary nodes are inconsistently visualized on mammography, we recently demonstrated a strong association between axillary metastases and fatty nodes in obese women on both MRI and mammography, despite non-visualized axillary nodes in 30% of obese women in that study [10].

Lymph nodes infiltrated by ectopic fat may be subject to organ dysfunction as has been shown in the liver, heart, pancreas, and skeletal muscle [21,22,23]. Obesity is associated with impairment of the immune and lymphatic system as evidenced by a higher risk of lymphedema and infections; however, the underlying mechanisms are not clearly understood [24]. Animal studies have shown that surgical disruption of the lymphatic system results in altered lipid trafficking, decreased lymphatic transport, and impaired immune cell function [25]. Importantly, the function of the immune system may be restored with weight loss in mice [26]. A recent study demonstrated a significant increase in the size of visceral lymph nodes in mice fed a high fat diet compared to mice fed a normal diet [27]. Although these findings have not been evaluated in humans, recent studies suggest that there is a likely correlation between lymphatic dysfunction and hyperlipidemia and the current gap in our understanding of this association has been identified as a major goal for future research [28,29]. Dietary cholesterols enter the lymphatic system of the bowel and are filtered through systemic lymph nodes before reaching the liver for clearance, a process known as reverse cholesterol transport [28,29]. We hypothesize that ectopic nodal fat may impair nodal function and thereby impact lipid transport due to nodal inflammation and dysfunctional lipid metabolism, mechanisms associated with organ dysfunction in other organs affected by ectopic fat deposition [3,4,5,6]. Fat infiltration of the lymph node hilum may further impair organelle function via structural changes of excess adipose tissue that may physically compress low-pressure lymphatic vessels. Adipose compression may decrease the normal egress of the lymph to the venous system thereby impairing normal trafficking of lipids through the lymphatics, a process similar to vascular compression described in renal sinus lipomatosis [30,31]. Weight loss associated with a decrease in nodal hilar fat may decrease the mechanical compression of the lymphatic vessels and thereby improve lymphatic flow through the lymph nodes leading to improved lipid clearance that contributes to the resolution of underlying dyslipidemia. The significance of patients without a change in fatty node size before and after bariatric surgery may reflect a subset of obese patients at risk for persistent dyslipidemia after weight loss compared to patients with decreased fatty node size. If confirmed in larger studies, this finding is consistent with prior studies showing that interventions that decrease ectopic fat are more strongly associated with improved cardio-metabolic health than interventions resulting only in weight loss and decreased BMI [32,33].

The limitations of our study include a moderate sample size and a single institution evaluation of patients with bariatric surgery and screening mammography. Furthermore, the mammographic evaluation of axillary lymph nodes is limited by the partial visualization of the inferior axilla that does not allow for a global mammographic visualization of the axillary nodes compared to a CT or an MRI. As a result, 39% of patients were excluded from our analysis due to partially or non-visualized axillary nodes, concordant with recently published rates of mammographically visible lymph nodes [20]. While we recently demonstrated a strong association between axillary metastases and fatty nodes on both MRI and mammography despite non-visualized axillary nodes in nearly 50% mammograms, it is nonetheless possible that limited visualization of the axillary lymph nodes may introduce bias when evaluating the association between fatty nodes and metabolic disease status. Future studies that include MRI-detected axillary nodes may address this limitation.

The strengths of our study include the exploration of a novel concept suggesting that changes in the size of fat-infiltrated axillary lymph nodes may reflect changes in lipid profiles after bariatric surgery. The high inter-rater agreement in the lymph node measurements shows that the evaluation of nodal size is highly reproducible. The majority of women in the United States undergo screening mammography but do not undergo screening for cardiovascular disease [34]. Despite limited mammographic lymph node visualization, the ability to leverage an existing screening study to opportunistically measure ectopic lymph node fat makes this technique potentially scalable and low cost. The primary innovative aspect of this project is our focus on fat deposition within human lymph nodes, organelles of the immune–lymphatic system tasked with lipid transport. Given the known association between organ dysfunction and increased cardiometabolic risk in other organs subjected to ectopic adipose deposition, fat-infiltration of the lymph nodes may similarly lead to immune–lymphatic dysfunction resulting in poor lipid transport and dyslipidemia. BMI has been criticized as a limited measure of cardio-metabolic risk, and it is well known that the distribution of body fat in ectopic locations has a stronger association with cardio-metabolic disease than BMI. Our preliminary study shows an association between the decreased size of fat-infiltrated nodes and the resolution of dyslipidemia after bariatric surgery. These preliminary findings raise the possibility that fat-infiltrated nodes may represent a new ectopic fatty depot. Further investigation is needed to confirm our findings and to further explore the association between fat-infiltrated lymph nodes and obesity-associated adverse health outcomes.

## Figures and Tables

**Figure 1 cells-11-00482-f001:**
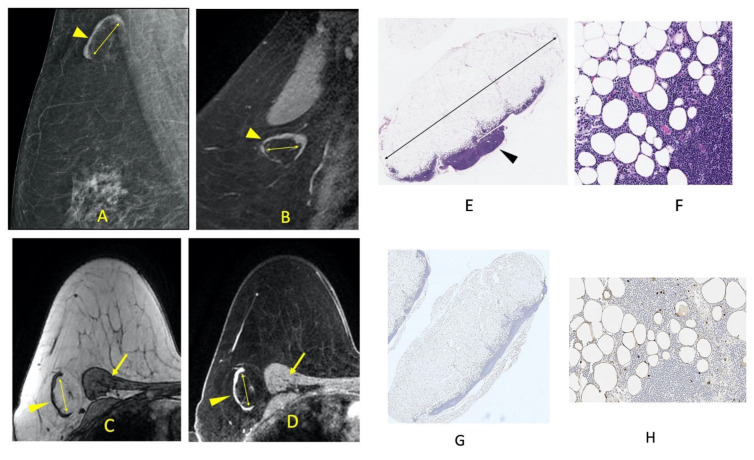
A non-metastatic fat-infiltrated axillary lymph node measuring 34 mm in a 67-year-old female with node-positive right breast cancer. Enlarged fatty hilum (double arrows) and thin peripheral cortex (arrowheads) are visualized on a right breast medio-lateral oblique mammogram (**A**), and on a sagittal contrast-enhanced right breast MRI (**B**). Axial breast MRI confirms that the expanded fatty hilum within the fat-infiltrated lymph node (double arrows) is composed of fat that is hyper-intense on the pre-fat-saturated sequence (**C**) and hypo-intense on the fat-saturated sequence (**D**). The fat signal within the lymph node hilum on the pre- and post-fat-saturated sequences is similar to the fat within the breast and axilla and is different from the peripheral thin lymph node cortex (arrowhead) and adjacent chest wall muscles (thin arrows). The histology of a non-metastatic sentinel node obtained at the time of breast cancer surgery confirms the extensive fatty expansion of the lymph node hilum (double arrow) with a slender rim of lymph node tissue retained at the periphery (arrowhead) (H&E, 1× magnification) (**E**). Sheets of adipocytes are seen infiltrating into and replacing the lymphocytes at a higher magnification (**F**) (H&E, 20× magnification). Low power microscopic image showing adipocytes replacing hilar and parenchymal lymph node tissue. The adipocytes are highlighted with a brown S100 immunostain (**G**) (S100, 1× magnification). Adipocytes, highlighted by expression of brown S100, seen infiltrating and replacing lymph node tissue. The background lymphocytes are S100 negative (**H**) (S100, 20× magnification).

**Figure 2 cells-11-00482-f002:**
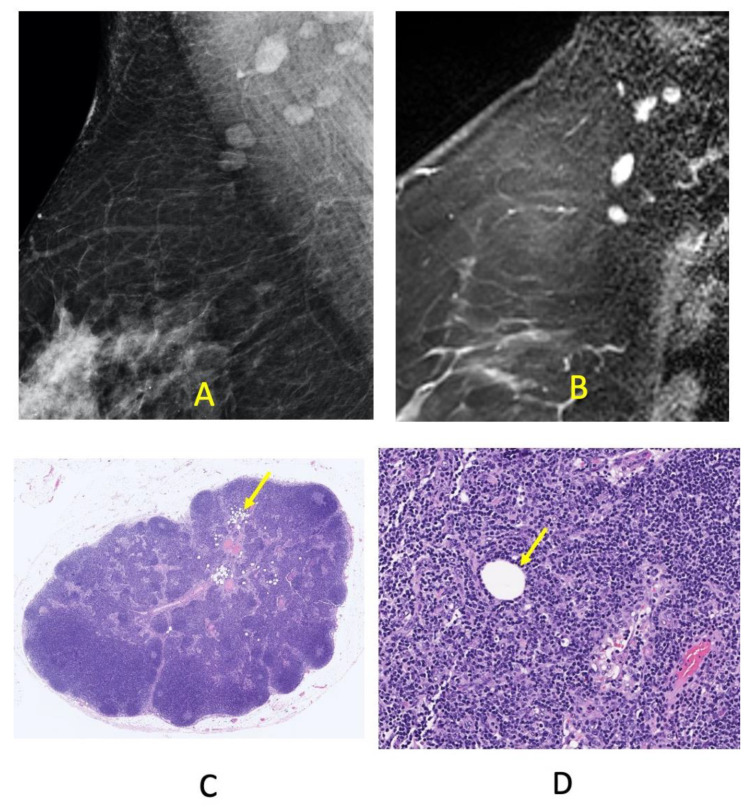
Normal right axillary lymph nodes, the largest measuring 13 mm, in a 62-year-old female with node negative right breast cancer. Normal nodes are visualized within the axilla on a right breast medio-lateral oblique mammogram (**A**), and on a sagittal contrast-enhanced right breast MRI (**B**). The histology of a sentinel node obtained at the time of breast cancer surgery confirms normal lymph node morphology with minimal fat. Rare patches of adipocytes are present; however, the lymph node architecture and cortical thickness are largely preserved (H&E, 1× magnification) (**C**). A single adipocyte is pictured within a background of lymphocytes and small capillaries (H&E, 20× magnification) (**D**).

**Figure 3 cells-11-00482-f003:**
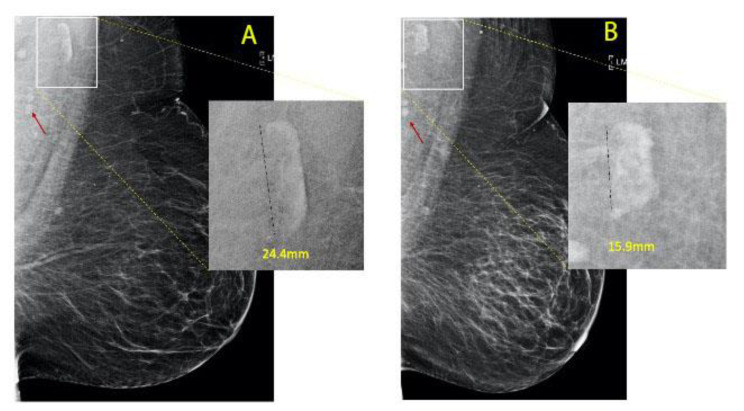
Left medio-lateral oblique mammogram before ((**A**), 2016) and after ((**B**), 2019) bariatric surgery. FIN (black dotted line) changed in size from 24.4 mm to 15.9 mm corresponding to a weight loss of 24.2 kgs. Normal lymph nodes for comparison ((**A**,**B**), red arrows). There is an associated decrease in the size of the right breast due to marked weight loss and decreased breast adipose tissue.

**Table 1 cells-11-00482-t001:** Study population characteristics between patients with no change and a change in the fat-infiltrated lymph node size. *p*-values for continuous and categorical variables were calculated by a two-tailed t-test and a two-tailed Fisher’s exact test, respectively. Weight is expressed in kilograms. Category “biguanide plus other” includes combination therapy with dipeptidyl peptidase-4 inhibitor, GLP-1 agonist, insulin, or sulfonylurea.

	Groups		
	No Change in FIN Size	Change in FIN Size	*p*-Value
*n*	39	45	
Age in years (mean (SD))	56.26 (7.50)	58.40 (7.10)	0.183
Pre-bypass weight (mean (SD))	117.0 (21.1)	122.6 (24.0)	0.264
Post-bypass weight (mean (SD))	91.3 (16.9)	89.5 (17.2)	0.647
Post-bypass change in weight (mean (SD))	25.71 (14.28)	33.02 (16.94)	0.037
Percentage of patients with T2DM (%)	28 (71.8)	33 (78.6)	0.654
Class of Antidiabetic Drug (%)			0.333
None	22 (56.4)	26 (59.1)	
Biguanide	8 (20.5)	3 (6.8)	
Biguanide plus other	4 (10.3)	9 (20.5)	
Insulin	4 (10.3)	4 (9.1)	
Sulfonylurea	1 (2.6)	2 (4.5)	
Number of patients with dyslipidemia (%)	31 (83.8)	30 (76.9)	0.644
Class of Antilipid Drug (N (%))			0.979
None	22 (56.4)	24 (54.5)	
Statin	16 (41.0)	19 (43.2)	
Statin, Fibric Acid	1 (2.6)	1 (2.3)	
Number of patients with hypertension (%)	29 (74.4)	30 (69.8)	0.829
Pre-bypass FIN size (mean (SD))	20.43 (6.17)	26.74 (8.81)	<0.001
Post-bypass FIN size (mean (SD))	18.65 (6.45)	20.37 (7.06)	0.252
Type of surgery (%)			1
Roux-en-y	21 (53.8)	24 (54.5)	
Sleeve gastrectomy	18 (46.2)	15 (38.5)	
Time to post-bypass mammogram (recorded in months; Mean (SD))	22.77 (16.58)	21.79 (13.52)	0.77

**Table 2 cells-11-00482-t002:** Mean pre-surgical lymph node size in patients with and without cardiometabolic disease. Mean pre-surgical lymph node size in millimeters was compared between individuals with and without hypertension, diabetes, dyslipidemia, or any combination of metabolic disease prior to surgery.

Metabolic Disease	Mean Pre-Surgical Lymph Node Size in Patients without Cardiometabolic Disease	Mean Pre-Surgical LN Size in Patients with Cardiometabolic Disease	*p*-Value
Hypertension	21.6	24.7	0.14
Diabetes	21.4	24.6	0.14
Dyslipidemia	22.5	24.1	0.5

**Table 3 cells-11-00482-t003:** Results of the multivariate linear model comparing the resolution of hypertension, diabetes, or dyslipidemia with a change in lymph node size. The effect size is represented by regression coefficients for continuous variables or by the adjusted odds ratios binary variables. Patients with a change in lymph node size were compared to patients without a change in node size, and patients who underwent a Roux-en-y gastric bypass surgery were compared to patients who underwent a sleeve gastrectomy. Weight is expressed in kilograms and lymph node size (LN) is expressed in millimeters.

	Effect Size *	Standard Error	*p*-Value
Resolution of hypertension			
Age	−0.07	0.05	0.18
Change in weight	0.04	0.03	0.14
Change in LN size	1.3 *	0.67	0.65
Surgery type	0.36	0.70	0.15
Resolution of DM			
Age	−0.044	0.05	0.41
Change in weight	0.05	0.04	0.52
Change in LN size	1.28	0.71	0.73
Surgery type	1.1	0.75	0.86
Resolution of dyslipidemia			
Age	−0.016	0.04	0.69
Change in weight	0.023	0.023	0.323
Change in LN size	5.5 *	0.63	0.006
Surgery type	1.25	0.65	0.73

Asterisk (*) represents exponentiated regression coefficient for binary variables.

## Data Availability

The data presented in this study are available on request from the corresponding author due to privacy issues.

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
