# Peer review of "Decrease in the Size of Fat-Enlarged Axillary Lymph Nodes and Serum Lipids after Bariatric Surgery"

_cells, 2022, doi:10.3390/cells11030482_

Round 1

Reviewer 1 Report

Reviewer’s comments:

The manuscript by Dwan et al; deals with an intriguing yet questionable concept of ectopic deposition of fat in the axillary lymph nodes of obese patients before and after gastric bypass surgery. The hypothesis of using fat infiltrated lymph node size changes as a marker for obesity associated adverse health out comes is interesting and refreshing. However, there are few major short comings that needs to be addressed thoroughly to establish this concept. My comments below.

  1. This study mainly depends on mammographically visualized fat infiltrated lymph nodes. The representative image provided by authors is not convincing enough to support all the statistical data provided based on those images. Fig1A, it is difficult to visualize the lymph nodes to come to a reasonable conclusion. It is surprising that out of 85 patients the authors were not able to find a more convincing image than the one provided. Previous works by the same group PMID: 34081259, 29144164 showed more convincing radiological images than the images provided here in this manuscript.

A better representative image should be provided possibly low magnification and zoomed in to show the fatty infiltration before and after the surgery.

  1. As authors mentioned in their discussion chronic obesity causes lymphatic dysfunction in mice and humans. But interestingly, in mice chronic obesity caused by high fat diet or genetic changes (Ob/Ob mice) show significant atrophy of lymph nodes with increased fat pad thickness around and adjacent to lymph nodes. To my knowledge, there is no literature to support the concept of fat accumulation in lymph nodes in obesity in mice. Lipid accumulation of lymph nodes is only reported in one transgenic mouse model where YAP/TAZ (hippo signaling) was knocked down in fibroblastic reticular cell (FRC) specifically (PMID: 31980640). The current group and few others previously showed fat infiltrated lymph nodes in humans through radiology techniques like mammography, CT and MRI to mention a few.

However, it will be nice if the authors show few images of obese or cancer patient lymph nodes for adipose tissue infiltration using basic histology or immunohistochemistry for lipid markers like Perilipin or BODIPY etc. this will help in establishing the concept of fat infiltrated lymph nodes without an iota of doubt. Such an evidence will be highly appreciated by non-clinical obesity and lymphatic researchers.        

Author Response

Reviewer #1

The manuscript by Dwan et al; deals with an intriguing yet questionable concept of ectopic deposition of fat in the axillary lymph nodes of obese patients before and after gastric bypass surgery. The hypothesis of using fat infiltrated lymph node size changes as a marker for obesity associated adverse health out comes is interesting and refreshing. However, there are few major short comings that needs to be addressed thoroughly to establish this concept. My comments below.

This study mainly depends on mammographically visualized fat infiltrated lymph nodes. The representative image provided by authors is not convincing enough to support all the statistical data provided based on those images. Fig1A, it is difficult to visualize the lymph nodes to come to a reasonable conclusion. It is surprising that out of 85 patients the authors were not able to find a more convincing image than the one provided. Previous works by the same group PMID: 34081259, 29144164 showed more convincing radiological images than the images provided here in this manuscript.

A better representative image should be provided possibly low magnification and zoomed in to show the fatty infiltration before and after the surgery.

We agree that it is difficult to portray the decrease in lymph node size with high quality images. We included a new example (Figure 3.) This shows the change in the morphology of the lymph node (although the change in size is not as dramatic as our original example), and includes magnified views to better illustrate nodal changes. Further, we added Figures 1 and 2 as suggested below, in order to provide references for the appearance of fatty and normal nodes on imaging and histology. These reference images may allow the reader to put the images of lymph node changes after gastric bypass into perspective. We would be happy to offer other possibilities for FIN change if this is not sufficient.

As authors mentioned in their discussion chronic obesity causes lymphatic dysfunction in mice and humans. But interestingly, in mice chronic obesity caused by high fat diet or genetic changes (Ob/Ob mice) show significant atrophy of lymph nodes with increased fat pad thickness around and adjacent to lymph nodes. To my knowledge, there is no literature to support the concept of fat accumulation in lymph nodes in obesity in mice. Lipid accumulation of lymph nodes is only reported in one transgenic mouse model where YAP/TAZ (hippo signaling) was knocked down in fibroblastic reticular cell (FRC) specifically (PMID: 31980640). The current group and few others previously showed fat infiltrated lymph nodes in humans through radiology techniques like mammography, CT and MRI to mention a few.

We very much appreciate the expert knowledge and literature cited. We read with interest that depletion of YAP/TAZ induces transition of fibroblast reticular cells into adipocytes in a transgenic mouse model  but is not associated with a change in lymph node size. We are aware of a recent publication (PMID: 31165249) showing significant increase in visceral lymph node size in mice after 13 weeks of high fat diet (associated with decrease in immune cell populations and proliferative ability) compared to mice fed a normal diet. We revised the discussion to include this reference.

However, it will be nice if the authors show few images of obese or cancer patient lymph nodes for adipose tissue infiltration using basic histology or immunohistochemistry for lipid markers like Perilipin or BODIPY etc. this will help in establishing the concept of fat infiltrated lymph nodes without an iota of doubt. Such an evidence will be highly appreciated by non-clinical obesity and lymphatic researchers.        In this retrospective analysis, the authors analyzed size of fat infiltrated lymph nodes in the axilla pre- and post gastric bypass surgery.

We thank the reviewer for this helpful comment and appreciate the opportunity to include additional images of fat-infiltrated nodes on mammogram, MRI, and histology to confirm that the expanded hilum is indeed secondary to fat deposition as shown in new Figure 1 compared to normal nodes in Figure 2. These images may help to lay the groundwork for morphologic changes we observe in fatty nodes and better inform the readers about our study aim. We thank the Reviewer for this suggestion.

Reviewer 2 Report

In this retrospective analysis, the authors analyzed size of fat infiltrated lymph nodes in the axilla pre- and post gastric bypass surgery. The aim of this study was to evaluate an association between weight loss and lymph node size as well as to assess the association between lymph node size and metabolic parameters. The authors included 156 patients in their study. In 45 patients, lymph node size decreased, and in 40 patients lymph node diameter did not change. The authors found, that patients with decreased lymph node size experienced a statistically better weight loss than patients with no change in size. Furthermore, the authors found that a decrease in lymph node size was associated to a resolution of dyslipidemia.

Please allow the following comments.

Introduction:

line 58 ff

If ectopic lymph node fat is likely to have adverse systemic events is a speculation to date. The hypothesis is plausible however not proven by the given quotes.

Possibly, the authors want to introduce the concept of metabolically healthy and metabolically unhealthy patients or individuals with obesity to their introduction, as they essentially describe this phenomenon.

Results :

The authors describe five different groups of patients. The two most relevant groups are patients with a change in lymph node size and patients with no change in lymph node size post gastric bypass surgery. For the 3 remaining groups, a determination of lymph node size was not possible due to different reasons. Therefore, I suggest mentioning the other groups only in a supplementary table as they do not give any added information. Instead, it would be very helpful for the reader to get a good description of group 1 (no change) and group 2 (change) in a separate table. Did these two groups differ in number of patients with type two diabetes, which anti-diabetic drugs were taken? Dis the groups differ in the number of patients with dyslipidemia, the number of patient with lipid lowering drugs these drugs also may have impact on fatty tissue deposition? I furthermore suggest a statistical comparison of group 1 and 2 with the hope of a better characterization of the two subgroups. One could hypothesize that a fatty infiltration of lymph nodes predominantly takes place in metabolically unhealthy individuals. Only when infiltrated, these lymph nodes also decrease in size. At least, that would be an interesting hypothesis to check.

If these two groups are compared, the analysis of weight loss should be adjusted to age, gender, presence of T2DM and initial weight as these are the known factors influencing weight loss after gastric bypass surgery. Also, information on the type of gastric bypass should be included. Not all gastric bypass surgery is alike. Were there patients with an OAGB? Was the biliopancreatic limb length identical in all patients?

At what time after surgery were the second mammograms taken? The authors mention a lower time limit of 6 months. Typically, patients experience ongoing weight loss up to one year after bariatric surgery. A mammogram taken six months after surgery may have been taken too early to detect the true effect. On the other hand, a mammogram taken five years after bariatric surgery when patients possibly have experienced weight regain, may underestimate the true effect as a fatty infiltration may have re-occurred.  Altogether, the bariatric cohort could be described in more detail.

Discussion:

The authors, at numerous times in the manuscript challenge the concept of BMI as a good measurement for patients with obesity. I fully agree with this, however when mentioned again in line 267 with the idea that ectopic fat deposition (…. in lymph nodes) is a better discriminator than BMI, this is a reach out far beyond the data foundation of this study.

Altogether, this is a very well written manuscript with a novel idea of describing and characterizing fat infiltrated lymph nodes pre- and post obesity surgery. However, the bariatric cohort should be described in more detail with a intergroup comparison of patients that present with decreased lymph node sizes and patients that do not.

Author Response

The aim of this study was to evaluate an association between weight loss and lymph node size as well as to assess the association between lymph node size and metabolic parameters. The authors included 156 patients in their study. In 45 patients, lymph node size decreased, and in 40 patients lymph node diameter did not change. The authors found, that patients with decreased lymph node size experienced a statistically better weight loss than patients with no change in size. Furthermore, the authors found that a decrease in lymph node size was associated to a resolution of dyslipidemia.

 Reviewer #2 

Please allow the following comments. 

Introduction: 

line 58 ff

If ectopic lymph node fat is likely to have adverse systemic events is a speculation to date. The hypothesis is plausible however not proven by the given quotes.

We thank the reviewer for keeping us in check. We corrected this statement to reflect that it is simply our hypothesis that nodal ectopic fat may have adverse effects, similar to changes of ectopic fat reported in other organ systems.

Possibly, the authors want to introduce the concept of metabolically healthy and metabolically unhealthy patients or individuals with obesity to their introduction, as they essentially describe this phenomenon.

Thank you for this helpful suggestion. We revised the introduction to include the concept of metabolically healthy obesity that is relevant to our study.

Results :

The authors describe five different groups of patients. The two most relevant groups are patients with a change in lymph node size and patients with no change in lymph node size post gastric bypass surgery. For the 3 remaining groups, a determination of lymph node size was not possible due to different reasons. Therefore, I suggest mentioning the other groups only in a supplementary table as they do not give any added information.

Thank you for helping us simplify the data display. A new table 1 is now included in the results section comparing the two relevant groups. The remaining three groups have been moved to Supplementary Table 1.

Instead, it would be very helpful for the reader to get a good description of group 1 (no change) and group 2 (change) in a separate table. Did these two groups differ in number of patients with type two diabetes, which anti-diabetic drugs were taken? Dis the groups differ in the number of patients with dyslipidemia, the number of patient with lipid lowering drugs these drugs also may have impact on fatty tissue deposition? I furthermore suggest a statistical comparison of group 1 and 2 with the hope of a better characterization of the two subgroups.

Thank you for highlighting the need to better describe the clinical characteristics of groups 1 and 2 and to further evaluate potential confounders in our study. We performed additional electronic chart review to collect data on DM, HTN, dyslipidemia and medications prescribed before gastric bypass. There were no significant differences in pre-bypass cardiometabolic disease or medications between the two groups. These are now included in Table 1.

One could hypothesize that a fatty infiltration of lymph nodes predominantly takes place in metabolically unhealthy individuals. Only when infiltrated, these lymph nodes also decrease in size. At least, that would be an interesting hypothesis to check.

This is indeed a very interesting idea and we thank the Reviewer for this excellent suggestion. We performed additional analysis that demonstrates that patients with hypertension, diabetes, or dyslipidemia do in fact have slightly larger lymph node size before surgery as suggested by this insightful comment. However, the difference was not significant in our study. Therefore we are not able to report a significant association between fat-infiltrated nodes and metabolically unhealthy individuals.  Nonetheless, these findings reflect a possible trend that may be significant in larger studies (Table 2).

If these two groups are compared, the analysis of weight loss should be adjusted to age, gender, presence of T2DM and initial weight as these are the known factors influencing weight loss after gastric bypass surgery.

Thank you for this suggestion. We created a revised Table 1 with these variables included.

Also, information on the type of gastric bypass should be included. Not all gastric bypass surgery is alike. Were there patients with an OAGB? Was the biliopancreatic limb length identical in all patients?

We thank the Reviewer for an interesting concept that we did not consider. We performed additional review of the electronic medical record to obtain this data. There was no significant difference in surgery type between the two relevant groups. This has been added to the results section and to revised Table 1.

Unfortunately, we were not able to identify the length of the biliopancreatic limb despite detailed review of the operative and clinical notes, and therefore we are not able to include this data.

This review did however allow us to identify a patient who was noted to have significant liver changes at the time of gastric bypass surgery, and a decision was made not to proceed with the Roux-en-y  procedure during surgery. She was a patient in the “FIN no-change” group, and she has now been removed from analysis resulting in 39 (previously 40) patients with no change in node size.  Results and tables have been adjusted to reflect this change.

At what time after surgery were the second mammograms taken? The authors mention a lower time limit of 6 months. Typically, patients experience ongoing weight loss up to one year after bariatric surgery. A mammogram taken six months after surgery may have been taken too early to detect the true effect. On the other hand, a mammogram taken five years after bariatric surgery when patients possibly have experienced weight regain, may underestimate the true effect as a fatty infiltration may have re-occurred.  Altogether, the bariatric cohort could be described in more detail.

Thank you for highlighting the need to report this relevant variable. We measured the number of months between the gastric bypass and the mammogram used for lymph node measurements and did not find a significant difference between the “FIN change” and “FIN no-change” groups (21.79 months s 22.79 months, p=0.77). This is now included in the results section and Table 1

Discussion:

The authors, at numerous times in the manuscript challenge the concept of BMI as a good measurement for patients with obesity. I fully agree with this, however when mentioned again in line 267 with the idea that ectopic fat deposition (…. in lymph nodes) is a better discriminator than BMI, this is a reach out far beyond the data foundation of this study.

We thank the reviewer for putting this into appropriate perspective. We revised the discussion to reflect the preliminary nature of our small study, and to highlight the need for larger studies to confirm our results, and to explore the association between fat-infiltrated nodes and obesity-associated cardiometabolic disease.

 Altogether, this is a very well written manuscript with a novel idea of describing and characterizing fat infiltrated lymph nodes pre- and post obesity surgery. However, the bariatric cohort should be described in more detail with a intergroup comparison of patients that present with decreased lymph node sizes and patients that do not.

Round 2

Reviewer 1 Report

The additional images improved the quality of manuscript. These new images now support the conclusions and the quantifications. The authors provided histological evidence of fatty deposition in lymph nodes. Even though I wont make it mandatory mandatory I still recommend the authors to perform some immunohistochemistry for adipocyte marker like perilipin or BODIPY staining to confirm the extent of fatty deposition in those lymph nodes. Considering the authors already have the tissue sections it is not a tough task to stain for adipocyte markers.  

Author Response

The additional images improved the quality of manuscript. These new images now support the conclusions and the quantifications. The authors provided histological evidence of fatty deposition in lymph nodes. Even though I wont make it mandatory mandatory I still recommend the authors to perform some immunohistochemistry for adipocyte marker like perilipin or BODIPY staining to confirm the extent of fatty deposition in those lymph nodes. Considering the authors already have the tissue sections it is not a tough task to stain for adipocyte markers.  

We reached out to our pathologist to request immunohistochemistry staining for adipocyte markers requested here. We unfortunately do not have perilipin or BODIPY immunostains, however the suggestion was to use S100, a cytoplasmic and nuclear stain that stains adipocytes. Those images are now included in revised Figure 1 (G and H).  Given the significant contribution of our pathologist to provide original H&E slides, further serve as a consultant for adipocyte IHC analysis, and authorize retrieval of nodal material for further analysis with S100, we are now including her as an author in this revised draft.

Reviewer 2 Report

Dear authors

Thank you very much for elaborately responding to all queries. Especially the new pictures and tables add to the understanding of the manuscript.

Please allow some additional comments for clarification:

The whole manuscript states, that gastric bypass surgery had been performed in these patients. In line 221 of the revised manuscript, the content now says that Roux-en Y gastric bypass OR sleeve gastrectomy had been performed in this cohort. Gastric bypass and Sleeve Gastrectomy are two very different bariatric operations. The type of surgery alone could have tremendous impact on weight loss, resolution of comorbidities and especially also lymph node size development. If truly sleeve gastrectomy and RYGB have been performed, this should be included in the analysis of table one and also should be included as a separate item in table 3. Also, the whole manuscript needed to be rewritten with the respect to gastric bypass wording. This needed to be changed to bariatric surgery or similar.

The comparison of metabolic disease and pre-bariatric surgery lymph node size is indeed interesting. Did you check, if there was a statistical difference when patients had hypertension or diabetes or dyslipidemia (all 3 groups combined)? Possibly, if you combine these three groups there may be a statistical difference.

In the initial review, reviewer #2 had to ask for an adjustment of weight loss analysis in table 1. In the answer to this query, you stated that the analysis in figure 1 was adjusted. Please clarify, did this adjustment reflect to “post bypass change in weight” ? In the text, there now is a section dealing with weight loss of the whole group, stating that only initial weight significantly predicted weight loss in the examined patient cohort. As this was the case in your data, possibly weight loss analysis in table 1 should only be adjusted to initial weight. For logical reasons, it makes sense to either adjust weight loss for the typically known confounders OR to adjust for the calculated confounders in particular data sets.

The new section of a multivariate regression model for weight loss of the whole group stands somewhat alone and may confuse the reader. To clarify, this was not the intention of the initial query regarding adjustment of weight loss analysis. Possibly, this was an editorial requirement.

For the final round of revisions, it would be easier to read if pictures, figures and figure legends would be presented on separate pages and not directly in the main text. Possibly again, this was an editorial requirement.

Author Response

This manuscript is a resubmission of an earlier submission. The following is a list of the peer review reports and author responses from that submission.

Round 1

Reviewer 1 Report

The hypothesis of the study is well explained and is based on solid physiopathological theories. The design of the study is adequate, but in methodology, the sample size calculation is missing. In results, is the decrease of 4.11 mm of FIN (lines 138 and 139) connected to the total study population (n=85)? If yes, the inclusion of a column in Table 2 demonstrating the characteristics of all studied patients could be necessary. Also, in Table 2, it is not clear the importance of the subgroups, once the authors did not discuss the changes in the FIN side and the outcomes of bariatric surgery. Did the authors measure the body composition? Changes in fat mass could be more reliable than BMI to explain differences in the size of fat-infiltrated axillary lymph nodes.

Author Response

Reviewer 1:

The hypothesis of the study is well explained and is based on solid physiopathological theories.

The design of the study is adequate, but in methodology, the sample size calculation is missing.

We thank Reviewer 1 for this comment. Given the exploratory nature of our study and the lack of established prior results, we performed a power analysis making use of our empirically observed sample statistics and demonstrated that our study was well-powered. We have added the following to the “Materials and Methods” section 2.4 of our manuscript lines 115-116: “Assuming a pooled standard deviation of 8.4 and per-group sample sizes of 40 and 45, there is a 93% power to detect a 6.4-mm change in LN at the P-value threshold of 0.05.”

In results, is the decrease of 4.11 mm of FIN (lines 138 and 139) connected to the total study population (n=85)? If yes, the inclusion of a column in Table 2 demonstrating the characteristics of all studied patients could be necessary.

Thank you for this comment. Table 1 has been modified to replace tables 1 and 2 in order to include all patients. Please note that due to formatting, it was not possible to save this as a tracked change so Table 1 is the tracked changes document is the NEW version. A mean decrease of 4.11 was observed in the subgroup of 85 patients who had measure-able, visible axillary lymph nodes before and after gastric bypass surgery, and the total study population consisted of 156 patients that are now also listed in modified Table 1.

Also, in Table 2, it is not clear the importance of the subgroups, once the authors did not discuss the changes in the FIN side and the outcomes of bariatric surgery.

Thank you for identifying this omission. Table 1 (originally tables 1 and 2)  now includes all patients.

Did the authors measure the body composition?

We are very interested in validated measures of body composition such as visceral adipose tissue (VAT) and muscle fat identified on CT scans; and we are particularly interested in the association between VAT and fat- infiltrated nodes. We are currently investigating the relationship between visceral fat and fatty nodes on PET-CT scans in women with breast cancer where we are able to image both visceral fat and axillary nodal fat and compare to outcomes. However, in this study population we do not have CT scans before and after bypass surgery. Therefore we used visible axillary nodes on screening mammograms.

We are Changes in fat mass could be more reliable than BMI to explain differences in the size of fat-infiltrated axillary lymph nodes

BMI is a measure of total body mass, and does not distinguish between fat, muscle, and bone mass. BMI is therefore not able to refine measures of body mass into measures of body composition that reflect the distribution of fat throughout the body and differentiate unhealthy ectopic fat from classic subcutaneous fat.  Ectopic fat depots are better predictors of cardio-metabolic disease than BMI, and may potentially represent targets for treatment and prevention. Research focused on body composition has emphasized the need to identify ectopic fatty depots and better understand the relationship between ectopic fat distribution and adverse health outcomes.

Reviewer 2 Report

In this study, the authors show first that fat-infiltrated axillary lymph nodes demonstrate a statistically significant decrease in size after gastric bypass surgery and second that the decrease in size is significantly associated with the resolution of dyslipidemia, independent of weight loss and patient age.

The authors present nodal fat deposition as a potential novel ectopic fatty depot associated with cardio-metabolic disease that could be evaluated to assess cardio-metabolic risk. These findings are novel and could have a potential clinical interest. The manuscript is concise (except for the discussion) and well written. The data are issued from the routine screening mammograms performed systematically before and after gastric bypass surgery. Consequently, the data are already available and quite easy to collect if the clinical interest is confirmed.

Comments and suggestions:

Abstract: In the abstract, the authors stated that they reviewed 156 patient files but only 85 could be considered for analysis. This should be mentioned.

Introduction :

The authors cite recent references in the text (4-5) before citing previous studies (8-9), it’s not consistent. That could be fixed by moving the sentence encompassing lines 42-44 right after reference 3.

Results:

-concerning the result presentation: to our opinion, it would be more consistent first, to exclude the groups and then, to present the group of patients who could be considered for analysis:

Initially 156 patients,

Then 31 excluded because no lymph nodes vizualised or normal lymph nodes, 40 excluded because they had no clearly visualized fat-expanded nodes on both pre and post-bypass surgery.

At the end: 85 were analysed (40+45)

-adipokines are discussed line 182. Is there any data available on adipokine serum levels in patients from the cohort?

-the Discussion section is too long when compared to the whole paper and should be shorten by 30%

Minor comments:

-Table 1: mix up between the groups with no change in FIN size and Change in FIN size?

-In the table 1: the term FIN is used for Fat-infiltrated lymph node whereas in the text, it is referred to fat-expanded lymph node. Please use the same term in text and table.

-Line 120: eight five instead of eighty five

Author Response

Reviewer 2:

In this study, the authors show first that fat-infiltrated axillary lymph nodes demonstrate a statistically significant decrease in size after gastric bypass surgery and second that the decrease in size is significantly associated with the resolution of dyslipidemia, independent of weight loss and patient age.

The authors present nodal fat deposition as a potential novel ectopic fatty depot associated with cardio-metabolic disease that could be evaluated to assess cardio-metabolic risk. These findings are novel and could have a potential clinical interest. The manuscript is concise (except for the discussion) and well written. The data are issued from the routine screening mammograms performed systematically before and after gastric bypass surgery. Consequently, the data are already available and quite easy to collect if the clinical interest is confirmed.

Comments and suggestions:

Abstract: In the abstract, the authors stated that they reviewed 156 patient files but only 85 could be considered for analysis. This should be mentioned.

Thank you for identifying this important clarification. We modified the abstract (line 15) and the results (lines 135-134) to reflect that 85 patients were included in the final analysis.

Introduction :

The authors cite recent references in the text (4-5) before citing previous studies (8-9), it’s not consistent. That could be fixed by moving the sentence encompassing lines 42-44 right after reference 3.

Thank you for pointing out this important chronologic oversight. We changed the order of the references to reflect the timing of the references

Results:

-concerning the result presentation: to our opinion, it would be more consistent first, to exclude the groups and then, to present the group of patients who could be considered for analysis:

Initially 156 patients,

Then 31 excluded because no lymph nodes vizualised or normal lymph nodes, 40 excluded because they had no clearly visualized fat-expanded nodes on both pre and post-bypass surgery.

At the end: 85 were analysed (40+45)

Thank you for the helpful suggestion regarding a clear way to present exclusion criteria. We have modified the results to reflect sequential exclusion and resulting 85 patients used for the analysis.

-adipokines are discussed line 182. Is there any data available on adipokine serum levels in patients from the cohort?

Unfortunately, we do not have adipokine data in this patient cohort. We are interested in adipokine profiles in obese women with fatty nodes and we are currently evaluating immune-histochemical profiles of normal and fat-infiltrated nodes that include adiponectin and leptin.

-the Discussion section is too long when compared to the whole paper and should be shorten by 30%

Thank you for this helpful feedback, the discussion has been reduced to 699 words from 1024 words

Minor comments:

-Table 1: mix up between the groups with no change in FIN size and Change in FIN size?

Thank you very much for pointing this out. After multiple iterations of the manuscript we flipped the table in error. We have corrected the table headings and adjusted with corrected data.

-In the table 1: the term FIN is used for Fat-infiltrated lymph node whereas in the text, it is referred to fat-expanded lymph node. Please use the same term in text and table.

Thank you for pointing out this inconsistency. We changed the term to fat-infiltrated nodes throughout the manuscript.

-Line 120: eight five instead of eighty five

Thank you for catching this typo—it has been corrected (currently line 134)

Reviewer 3 Report

Ectopic fat deposition of obesity is associated with organ dysfunction. Fat deposition within the lymphatic system and lymphatic dysfunction among the obese population is a topic poorly investigated. Dawn and colleagues have evaluated axillary lymph node size, patient weight in pounds, body mass index, and a diagnosis of hypertension, type 2 diabetes, and dyslipidemia in one hundred fifty-six women who underwent routine screening mammograms before and after gastric bypass surgery. They observed that fat-infiltrated axillary lymph nodes demonstrated a statistically significant decrease in size after gastric bypass surgery with a mean decrease of 4.11 millimeters. They concluded that decrease in lymph node size was significantly associated with resolution of dyslipidemia, independent of weight loss and patient age.

A complete panel of metabolites before and after surgery, should be included in the manuscript (insulin, glucose, TG, CHL, LDL-CHL, HDL-CHL…).

Please described race of women and its impact on multivariate analyses.

A multicentric study would improve the significance and relevance of findings.

From a research point of view, the study is novel but very descriptive. The manuscript would be substantially improved by showing if inflammatory markers of immune cells have changed (e.g. T-lymphocytes, B-lymphocytes, dendritic cells,…). Likewise, a panel of pro-inflammatory cytokines).

If possible, biopsies of axillary lymphoid size would be very informative looking at inflammatory markers.

What is the significance of patients with fat-expanded modes on both pre- and post-bypass surgery? Any explanation?

What is the relevance of findings if about 25% (n=40) of patients were excluded because fat-expanded modes on both pre- and post-bypass surgery; no lymph nodes visualized in the 126 axilla (n = 21,13%) or normal lymph nodes (n = 10, 6.4%). So, basically, about 50% of patients were excluded due to lymph node size measurements issues. Is this significant? Is not the study biased?

Author Response

Reviewer 3:

Ectopic fat deposition of obesity is associated with organ dysfunction. Fat deposition within the lymphatic system and lymphatic dysfunction among the obese population is a topic poorly investigated. Dawn and colleagues have evaluated axillary lymph node size, patient weight in pounds, body mass index, and a diagnosis of hypertension, type 2 diabetes, and dyslipidemia in one hundred fifty-six women who underwent routine screening mammograms before and after gastric bypass surgery. They observed that fat-infiltrated axillary lymph nodes demonstrated a statistically significant decrease in size after gastric bypass surgery with a mean decrease of 4.11 millimeters. They concluded that decrease in lymph node size was significantly associated with resolution of dyslipidemia, independent of weight loss and patient age.

A complete panel of metabolites before and after surgery, should be included in the manuscript (insulin, glucose, TG, CHL, LDL-CHL, HDL-CHL…).

Thank you for this insightful comment. We created an additional table to address changes in metabolic panel before and after surgery, and due to the large size of this table, we included this file as Supplementary Table 1, submitted at the end of the maunscript. Supplementary Table 1 is also mentioned in the results section (line 144). We additionally modified Table 1 in the main draft to include a more comprehensive overview of clinical and metabolic characteristics before and after surgery

Please described race of women and its impact on multivariate analyses.

The cohort for this study is white, concordant with our rural geographic location and catchment area. We have added this to the materials and methods as this does limit the generalizability of our preliminary study.

A multicentric study would improve the significance and relevance of findings.

Yes we absolutely agree. This is a preliminary study in which we evaluated all available obese women who had gastric bypass in our institution since 2011 when our current electronic medical record was instituted (n=731). We would like to expand this research to multiple sites to increase the diversity of the patient population and the sample size.

From a research point of view, the study is novel but very descriptive. The manuscript would be substantially improved by showing if inflammatory markers of immune cells have changed (e.g. T-lymphocytes, B-lymphocytes, dendritic cells,…).. Likewise, a panel of pro-inflammatory cytokines).

If possible, biopsies of axillary lymphoid size would be very informative looking at inflammatory markers.

Yes we agree that serum markers and nodal biopsies would be very informative however they were not available in this patient cohort.

We are however currently investigating markers of T and B cell function, inflammation (TNFa and IL6), adipokines, (adiponectin and leptin), and lipid metabolism (lipoprotein lipase, and fatty acid synthase) with immuno-histochemical analysis of the sentinel lymph nodes of obese women with breast cancer in whom we found a strong association between enlarged fat-infiltrated lymph nodes and nodal metastases when adjusted for clinical and tumor characteristics. We do not have preliminary data to share at this time. Unfortunately, we did not have serum markers or nodal tissue for analysis in this gastric bypass cohort

What is the significance of patients with fat-expanded modes on both pre- and post-bypass surgery? Any explanation?

Our current hypothesis is that lymph nodes infiltrated by fat may be subject to similar ectopic adipose-induced dysfunction identified in other organs infiltrated by fat such as the liver and skeletal muscle. In the lymphatic system, ectopic fat may affect nodal function via inflammation that induces cortical dysfunction, as well as compression of hilar vessels and efferent lymphatic channels that prevent adequate lymphatic flow out of the lymph node to join the venous system via the thoracic duct. Both of these mechanisms may impair the role of the lymphatics in lipid clearance. In particular, significant post-bypass surgery weight loss may diminish hilar fat in the lymph nodes and relieve compression on the efferent lymphatics thereby improving lipid clearance.

What is the relevance of findings if about 25% (n=40) of patients were excluded because fat-expanded modes on both pre- and post-bypass surgery; no lymph nodes visualized in the 126 axilla (n = 21,13%) or normal lymph nodes (n = 10, 6.4%). So, basically, about 50% of patients were excluded due to lymph node size measurements issues. Is this significant? Is not the study biased?

This is a very interesting question and we appreciate the opportunity to discuss lymph node visibility. Reported mammographic visibility of axillary nodes among all women ranges from 50-78% due variable patient mobility, mammographic technique (positioning and compression), and anatomic variation in node location, size and number. Our results are within this range: 80% women had axillary nodes visible on pre or post bypass surgery mammograms while 54% women had visible nodes on both pre and post bypass mammograms. In a recent study evaluating fatty nodes in obese women with breast cancer, we measured fat-infiltrated nodes on mammograms and breast MRI. We found that non-visualized lymph nodes on mammograms were associated with exams performed at outside institutions. This finding supports the role of mammographic technique in visibility of axillary nodes (positioning and degree of compression).

We further found that breast MRI affords visualization of a larger number of axillary nodes compared to mammography due to a larger field of view that includes level 1, 2, and 3 axillary nodes compared to partial visualization of level 1 axillary nodes on mammograms. The mean size of MRI-detected nodes was significantly larger than mammographically-detected nodes and was due to larger size and increased fat within deep level 1 and 2 nodes compared to more superficial mammographically visualized level 1 nodes. Nonetheless, both mammographic and MRI-detected fatty nodes were independently associated with higher likelihood of nodal metastases.

Importantly, with breast MRI for comparison, we did not find any significant differences among women with visualized and non-visualized axillary nodes and breast cancer outcomes. The mechanisms responsible for nodal metastases and metabolic changes are presumably different (although there may be some overlap), and we therefore cannot exclude the possibility that non-visualized nodes bias our results. Future studies may benefit from availability of MRI or CT to better visualize changes in lymph node size in patients with significant weight loss.

Round 2

Reviewer 3 Report

Thank you very much for addressing reviewer´s concerns.